# Superconducting diode effect in a meso-wedge geometry with Abrikosov vortices.

C. A. Aguirre[1,2] J. Barba-Ortega[3] A. S. de Arruda[1] and J. Faúndez[4]

[1] *Instituto de Física, Universidade Federal de Mato-Grosso, Cuiabá, Brasil.*
[2] *Escuela Superior de Empresa Ingeniería y Tecnología, Bogotá, Colombia.*
[3] *Departamento de Física, Universidad Nacional de Colombia, Bogotá, Colombia. and*
[4] *Departamento de Física y Astronomía, Universidad Andres Bello, Santiago 837-0136, Chile.*
(Dated: December 9, 2025)

In this study, we explore the behavior of a superconducting meso-wedge geometry in 3+1 dimensions (three spatial dimensions plus time) subjected to external transport currents at its boundaries and surfaces, as well as external fields applied along the $\hat{z}$-direction. The transport currents are included as two opposite polarities, $\mathbf{J} > 0$ and $\mathbf{J} < 0$, respectively. Using the generalized time-dependent Ginzburg-Landau theory and considering the order parameter $\kappa$, we focus on two scenarios: a fixed external magnetic field with variable $\kappa$, and fixed $\kappa$ with variable external magnetic field. As a result, under both scenarios, we analyze the voltage-current characteristics of the superconducting meso-wedge, finding that the critical currents differ between polarities, demonstrating the system's non-reciprocity. We further examine the efficiency of the diode as a function of $\kappa$ and the external magnetic field applied. Furthermore, our observations reveal that the current polarity strongly influences the vortex configuration, the parameter $\kappa$, and the applied magnetic field. In particular, the formation of Abrikosov-type vortices exhibits pronounced inhomogeneity depending on the direction of the transport currents. This underscores that the diode effect in the superconducting meso-wedge is intimately associated with the anisotropic nucleation of Abrikosov vortices. Notably, the emergence of polarity-dependent vortex patterns can serve as a distinctive hallmark of the diode effect in these superconducting meso-wedge geometries.

## I. INTRODUCTION

The study of Abrikosov vortices in superconducting systems and their relationship to the diode effect has become a central topic in contemporary condensed matter physics [1–3]. Abrikosov vortices arise in type II superconductors, which are characterized by the Ginzburg–Landau parameter (GLP) $(\kappa > 1/\sqrt{2})$. These vortices represent quantized magnetic flux lines that penetrate the superconductor under an external applied magnetic field, allowing the coexistence of superconducting and normal-state regions [4–6]. Physically, an Abrikosov vortex consists of a core where the superconducting order parameter $\psi$ vanishes, surrounded by circulating super-currents (Meissner currents) that screen the magnetic field. The arrangement and dynamics of these vortices are governed by a balance of long-range attractive and short-range repulsive interactions, strongly influenced by the geometry of the sample and external conditions [7–12].

A powerful theoretical framework to describe these phenomena is the Ginzburg–Landau theory (GL), which provides a macroscopic description of superconductivity in terms of a complex order parameter $\psi$ and accounts for the interplay between magnetic fields, currents, and the superconducting condensate. In recent years, extensions of the GL theory have enabled the exploration of more complex systems, including mesoscopic superconductors, multi-band and multi-component superconductors, fractional vortices, and topological phases [13–16]. Importantly, the interaction of vortices with system geometry can generate rich spatial patterns that critically impact the electromagnetic and transport properties of superconducting materials [17]. Geometric confinement can alter the mobility, stability, and configurations of the vortex, thus tuning the macroscopic response of the material [18–20].In this context, Abrikosov vortices provide a natural platform to explore the superconducting diode effect. Their cores and circulating currents generate strongly inhomogeneous current patterns that are coupled to the geometry and boundaries of the sample, thus introducing an intrinsic directionality in the transport response [21–24]. As a result, mesoscopic superconductors hosting vortices can display distinct critical currents for opposite current directions, making them ideal systems to investigate and enhance diode-like superconducting behavior [25–28].

A related and rapidly emerging phenomenon is the superconducting diode effect, which refers to the asymmetric response of a superconductor to transport currents of opposite polarities [29–31]. This effect manifests itself as a directional dependence in the critical current or resistance, enabling superconducting transport in one direction while suppressing it in the opposite direction. The diode effect is generally associated with symmetry-breaking mechanisms, such as geometric asymmetry, intrinsic material anisotropy; or, in some cases, due to external applied magnetic fields, which break time-reversal symmetry [31]. While the diode effect has been widely studied in Josephson junctions [32–37] and engineered hetero-structures, its manifestation in mesoscopic superconductors, particularly in systems hosting Abrikosov vortices, remains relatively unexplored. Although currently, the efficiency of the diode effect has been explored in different configurations [38, 39] and even accompanied by altermagnets [40] and in some studies, where the efficiency peaks around 20-30% [41–43]. Anisotropy and geometric asymmetry can strongly modify vortex

behavior, introducing directionality into the superconducting response and enabling novel functionalities for electronic applications [29, 31]. Theoretical approaches such as the London model and numerical simulations based on the time-dependent Ginzburg–Landau equations have been employed to investigate vortex–geometry interactions [18, 19]. These studies have revealed mechanisms by which the interplay between vortices and asymmetric boundaries can induce non-reciprocal current transport, providing a pathway to superconducting rectification and new device concepts. Experimental observations have demonstrated the diode effect in a range of systems, including van der Waals hetero-structures without magnetic fields [38, 44–46], thin films of conventional superconductors under weak fields [47–50], and twisted tri-layer graphene, where the coexistence of superconductivity and magnetism enables diode-like behavior [51, 52]. A central requirement for the realization of the superconducting diode effect is the simultaneous breaking of fundamental symmetries. As emphasized in Ref. [53], nonreciprocal supercurrents emerge only when both inversion symmetry ($\mathscr{P}$) and time-reversal symmetry ($\mathscr{T}$) are broken. Inversion symmetry can be lifted by structural asymmetry or by intrinsic spin–orbit interactions such as those of Rashba type, whereas time-reversal symmetry is broken by the presence of an external magnetic field or magnetic order. The concomitant violation of $\mathscr{P}$ and $\mathscr{T}$ generates magnetochiral anisotropy, which manifests as direction-dependent critical currents, $J_c^+ \neq J_c^-$. Within this symmetry-based framework, the meso-wedge geometry provides a natural platform to realize the diode effect: the wedge shape intrinsically breaks inversion symmetry through its triangular confinement, while the penetration of Abrikosov vortices under an applied magnetic field explicitly breaks time-reversal symmetry. The cooperative action of geometry and vortex dynamics therefore creates the fundamental conditions for nonreciprocal superconducting transport, as recently confirmed by experimental observations in wedge-shaped mesoscopic superconductors [54].

In this work, we investigate the superconducting diode effect in a meso-wedge-shaped superconductor with inversion symmetry breaking and time reversal symmetry breaking given by an applied magnetic field along the $\hat{z}$-direction, see Fig. 1(a) for details. Our main goal is understanding how geometric anisotropy influences these system's critical currents, diode efficiency, and Abrikosov vortices nucleation. Specifically, we analyze the behavior of the superconducting diode effect as a function GLP, labeled $\kappa$, and the applied external magnetic field, $\mathbf{H}_z$. We provide a detailed explanation of how periodic energy barriers at the system boundaries contribute to the rectification effect, distinguishing this mechanism from that of conventional Josephson-based superconducting diodes [55]. As a result, we show that the meso-wedge geometry, which breaks inversion symmetry, together with the presence of the magnetic field, which breaks time-reversal symmetry, leads to non-homogeneous critical currents of opposite polarity, $\mathbf{J}_c^+ \neq \mathbf{J}_c^+$. This asymmetry manifests as a finite superconducting-diode efficiency, which

depends on the parameter $\kappa$ and on the intensity of $\mathbf{H}_z$. Moreover, the diode effect is accompanied by asymmetric patterns in the nucleation of Abrikosov vortices.

This paper is organized as follows. Section II describes the theoretical framework, including the time-dependent Ginzburg–Landau equations used in our analysis. Section III shows the main results from our numerical simulations, including voltage-current characteristics, first critical current, diode efficiency, and vortex nucleation varying $\kappa$ or magnetic fields. Finally, we summarize our conclusions and outline future directions in Section IV.

## II. THEORY AND MODEL

We studied a real three-dimensional superconducting meso-wedge under a fixed external magnetic field ($\mathbf{H} = \mathbf{H}_z$) applied in $\hat{z}$ direction. The geometry of our superconducting meso-wedge is illustrated in Fig. 1 (a)-(b). The superconducting meso-wedge fills the domain $\Omega$. The interface between the lateral region and the vacuum is denoted by $\partial\Omega_i$, $i = 1, 2$. The dimension of the numerical sample is A×B×C. With this in mind, an external transport current ($\mathbf{J}$) -in $-\hat{x}$-direction- is applied to the superconducting meso-wedge on the lateral of the geometry, see Fig. 1(a). In this work, we employ the Generalized Time-Dependent Ginzburg-Landau Theory (TDGL), formulated under the dirty limit and expressed in dimensionless units, as described in Refs. [56–60].

$$\frac{1}{\sqrt{1+\Gamma^2|\psi|^2}}\left[\frac{\partial\psi}{\partial t} + \frac{\Gamma^2}{2}\frac{\partial|\psi|^2}{\partial t} + \Phi\psi\right]$$
$$= (i\nabla + \mathbf{A})^2\psi + \psi(1 - |\psi|^2), \quad (1)$$

for the potential vector:

$$\frac{\partial\mathbf{A}}{\partial t} = \mathbf{J} - \kappa^2(\nabla \times \nabla \times \mathbf{A}), \quad (2)$$

where

$$\mathbf{J} = \mathrm{Re}\left[\bar{\psi}(-i\nabla - \mathbf{A})\psi\right] - \nabla\Phi, \quad (3)$$

and $\kappa$ characterizes the type of superconductor by relating the spatial variation of the order parameter $\psi$ to the magnetic field penetration into the sample. In GL theory, $\kappa$ is a dimensionless quantity defined as the ratio between the magnetic penetration depth and the coherence length of the superconductor. This parameter determines whether a material is classified as type I or type II: for $\kappa < 1/\sqrt{2}$ the superconductor is type I, while for $\kappa > 1/\sqrt{2}$ it is type II. Importantly, we associate the mechanical rigidity of the superconducting vortex lattice with $\kappa$. Larger values of $\kappa$ correspond to a softer vortex lattice, thereby facilitating vortex penetration at lower values of $\mathbf{J}$'s. In conjunction with the continuity equation, which also adopted the Coulomb gauge $\nabla \cdot \mathbf{A} = 0$ and Maxwell's first law, the expression for the

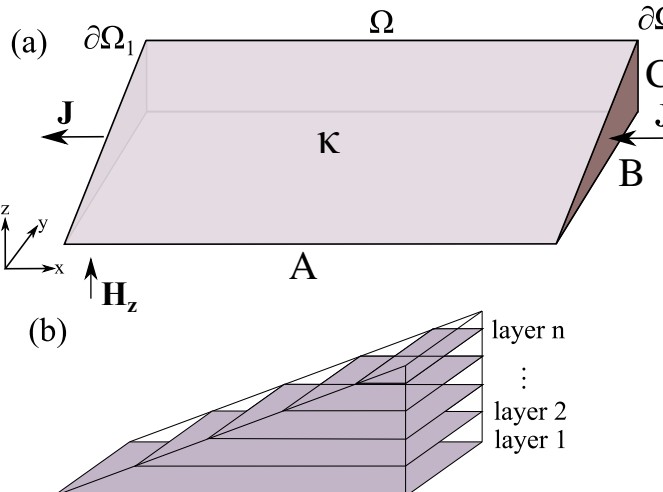

FIG. 1. (a) Schematic representation of the superconducting meso-wedge geometry. The dimensions of the external numerical mesh are A= 30$\xi$, B=C= 15$\xi$. (b) Real three-dimensional superconducting sample with meso-wedge geometry and its projection of $n$-layers for the superconducting meso-wedge sample. The inclusion of external/transport currents (**J**) -in $-\hat{x}$-direction- is given by the lateral faces $\partial\Omega_i$, $i = 1, 2$, the external magnetic field (**H** = **H**$_z$) is in the $\hat{z}$-direction.

scalar potential ($\Phi$) is obtained as a Poisson time-dependent equation, which is given by:

$$\nabla^2\Phi = \frac{\partial\rho}{\partial t} = -\nabla \cdot \mathbf{J}. \tag{4}$$

The Eqs. (1)-(4) are solved in a self-consistent approach and the Neumann boundary conditions for the potential/external current are $\hat{n} \cdot \nabla\Phi = -\mathbf{J}$ in sections with external current ($\partial\Omega_i$, $i = 1, 2$) and $\hat{n} \cdot \nabla\Phi = 0$ in the other sections, with $\hat{n}$ being a surface normal vector. In addition, $\mathbf{J}$ corresponds to the total superconducting current density (Meissner plus external/transport).

In the quasi-steady regime, $\partial_t\rho \simeq 0$, so that Laplace's equation ($\nabla^2\Phi = 0$) holds in the bulk, with solvability requiring that the net injected current equals the extracted one. Because a pure Neumann problem determines $\Phi$ only up to an additive constant, we fix the reference by enforcing, e.g., $\langle\Phi\rangle_{\partial\Omega_2} = 0$. The instantaneous voltage is then computed as the difference of surface-averaged potentials at the electrodes (leads),

$$V(t) = \langle\Phi\rangle_{\partial\Omega_1} - \langle\Phi\rangle_{\partial\Omega_2}, \qquad \langle\Phi\rangle_S = \frac{1}{|S|}\int_S \Phi dS, \tag{5}$$

where, $S$ denotes the leads surface—i.e., a subset of the boundary $\partial\Omega$—over which the potential is averaged. The voltage follows from time-averaging $V(t)$ over the stationary window. Furthermore, in Eqs. (1), (2), and (3), dimensionless units were introduced as follows: the order parameter $\psi$ is in units of $\psi_\infty = \sqrt{-\alpha/\beta}$ (the order parameter at the Meissner-Oschenfeld state), where $\alpha$ and $\beta$ are two phenomenological

constants; $H_1$ is the first critical field (Meissner-Oschenfeld field); lengths are in units of the coherence length $\xi$; time is in units of the Ginzburg-Landau characteristic time $t_{GL} = \pi\hbar/(8K_BT_c)$; fields are in units of $H_{c2}$, where $H_{c2}$ is the bulk second critical field; the vector potential $\mathbf{A}$ is in units of $\xi H_{c2}$; $\kappa = \lambda/\xi$ is the GLP, which describes the type of superconductor as a function of the spatial variation of the order parameter $\psi$ and the penetration of the magnetic field into the sample and $\Gamma = 10$. In addition, we use the triple convergence rule for time [61, 62].

$$dt_1 = \frac{a\eta}{4\sqrt{1+\Gamma^2}}, \quad dt_2 = \frac{a\beta}{4\kappa^2}, \quad dt_3 = \frac{a\nu}{4\zeta^2}, \tag{6}$$

and

$$\Delta t \leq min\{dt_1, dt_2, dt_3\}, \quad a^2 = \frac{2}{\frac{1}{\delta x^2} + \frac{1}{\delta y^2} + \frac{1}{\delta z^2}}. \tag{7}$$

For numerical calculations, we use the mesh size $\delta x = \delta y = \delta z = 0.1$, the constant values: $\eta = 5.79$, $\beta = 1.0$, $\zeta = 0.50$, $\kappa$ will be variable in a section of this manuscript, and $\nu = 0.03$ [60]. For tolerance in convergence of the order parameter $\psi$, we employ $\varepsilon = 1.0^{-9}$, and the errors are of order $O(\Delta x)^2$ for space and time. For boundary conditions of the order parameter, we employ Robin's boundary condition $\mathbf{n} \cdot (i\nabla + \mathbf{A})\psi = -i\psi/b$, with $\mathbf{n}$ being a surface normal vector and $b$, the de-Gennes extrapolation parameter and we have taken $b \to \infty$ in the lateral contacts and $b = 0$ in the rest of the sample. Finally, as shown in Fig. 1(a), the dimensions of the external numerical mesh are $A = 30\xi$, $B = C = 15\xi$, and in Fig. 1(b), the superconducting meso-edge has $n$-layers, where C=$n\xi$.

## III. NUMERICAL RESULTS

### A. Current, non-reciprocity and efficiencies

This section presents the numerical results obtained for the superconducting meso-wedge. To this end, we begin by showing in Fig. 2 the voltage response ($V$) as a function of the externally applied transport current (**J**) for different values of the GLP ($\kappa$) and the fixed magnetic field, **H**$_z$ = 1.0. It can be observed that the first critical currents, $\mathbf{J}_c$, for positive and negative current directions (**J** > 0 and **J** < 0) are not identical. In particular, the asymmetry between the two critical values, $\mathbf{J}_c^+ \neq \mathbf{J}_c^-$—where "+" corresponds to **J** > 0 and "–" to **J** < 0—constitutes the basis of the superconducting diode effect. This effect arises because vortex motion is triggered at different threshold currents depending on the direction of **J**. Also, the values of $\kappa$ considered are all within the type-II superconducting regime and are ordered in increasing magnitude. As $\kappa$ increases, we observe a reduction in the $\mathbf{J}_c's$. This behavior arises from an increase in penetration depth, which modifies the slope of the voltage-current curve and leads to the emergence of a transient resistive state. This state is associated with the nucleation and motion of vortices inside the superconducting meso-wedge. Moreover,

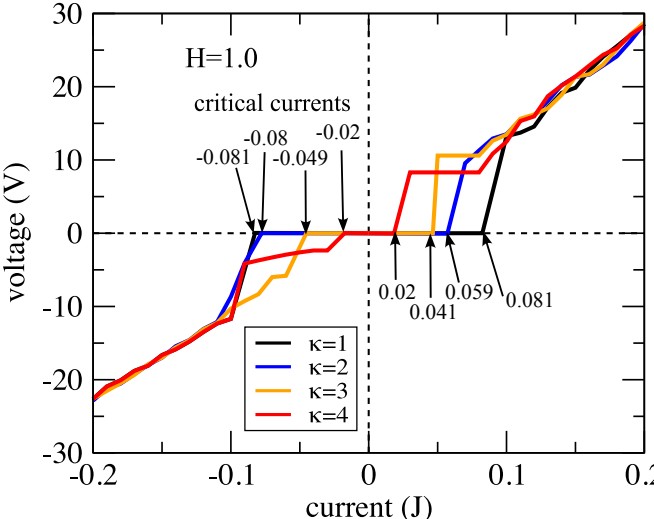

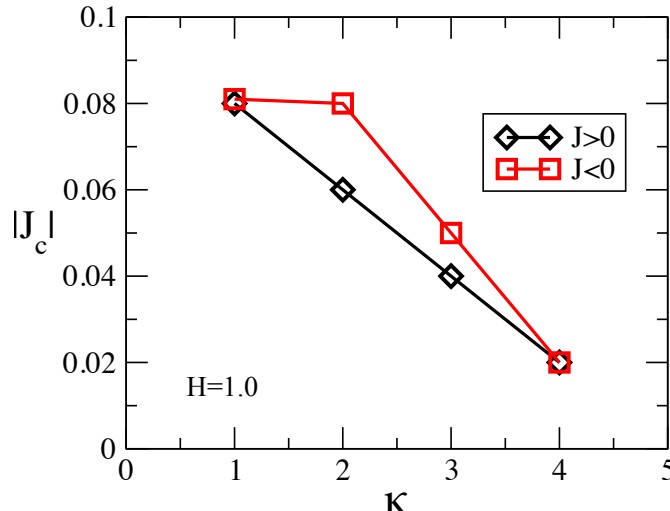

FIG. 2. Voltage response ($V$) as a function of the transport current ($\mathbf{J}$) for a fixed external magnetic field applied along the $z$-direction, $\mathbf{H}_z = 1.0$, and for both current polarities ($\mathbf{J} > 0$ and $\mathbf{J} < 0$). The results are shown for different values of the Ginzburg–Landau parameter, $\kappa$.

FIG. 3. Magnitude of the first critical current ($\mathbf{J}_c$) for both polarities of the transport current ($\mathbf{J_c}$), plotted as a function of Ginzburg-Landau parameter, $\kappa$. Results are shown for fixed values of the external magnetic field, $\mathbf{H} = 1.0$, for the superconducting meso-wedge sample.

the transport current along the $-\hat{x}$-direction generates a preferred direction for vortex movement, driven by the Lorentz force [19]. The resulting voltage jumps—reminiscent of the Shapiro steps [63]—are related to the maximum velocity of the vortices, given by $v^* = V/B$, where $B$ is the magnetic flux density. The vortex velocity is also influenced by vortex–vortex interactions, which depend on the Meissner current's circulation. These interactions can either enhance or suppress the vortex mobility. Therefore, we associate the mechanical rigidity of the superconducting vortex lattice with the parameter $\kappa$ (here referred to as the GLP): larger values of $\kappa$ reduce the lattice stiffness, facilitating vortex entry at lower values of the external transport current.

Fig. 3 shows the magnitude of the first critical currents ($\mathbf{J_c}$) -the onset of resistive states- as a function of $\kappa$ for both the polarities of $\mathbf{J}$ and fixed $\mathbf{H_z} = 1.0$. It is important to note that the first $\mathbf{J_c}$ corresponds to the threshold current above which vortices, either kinematic or Abrikosov, become depinned and start to move across the meso-wedge. At this point, the Lorentz force surpasses the pinning force, resulting in vortex motion, energy dissipation, and a measurable variation in the scalar potential. This regime therefore marks the onset of the resistive state within the mixed phase between the superconducting and normal states. For completeness, we note that a second $\mathbf{J_c}$ also exists. However, this regime is not considered in the present study, as the superconducting state is completely destroyed due to Cooper-pair breaking, and the system transitions entirely to the normal state.

Taking the above into account, when we consider $\kappa = 1.0$, only a slight difference is observed between $\mathbf{J_c}'s$, which may be attributed to the high rigidity of the superconducting condensate, where the strong coupling between Cooper

pairs suppresses the vortex dynamics. In contrast, for $\kappa = 2.0$, a more pronounced asymmetry emerges, suggesting that a lower rigidity of the vortex lattice facilitates vortex motion and optical phonon-like oscillations. This asymmetry increases with $\kappa$ up to a certain value, after which it decreases for higher values of $\kappa$. Beyond the $\mathbf{J_c}$ threshold, the mesoscopic superconducting wedge exhibits a nearly Ohmic (linear) response. Importantly, the observed difference between the $\mathbf{J_c}'s$ for the positive and negative polarities of $\mathbf{J}$ confirms the presence of the superconducting diode effect. This asymmetry reflects the breakup of the spatial inversion symmetry in the superconducting meso-wedge [63].

We now proceed to characterize the diode effect quantitatively. To this end, it is useful to define a signed efficiency parameter, following the approach introduced in Ref. [64] (and references therein), which quantifies the degree of rectification by measuring the asymmetry between the critical currents for opposite current polarities. This efficiency parameter is defined as:

$$\gamma_d(\mathbf{H}) = \frac{|J_c^+(\mathbf{H}) - |J_c^-(\mathbf{H})||}{J_c^+(\mathbf{H}) + |J_c^-|(\mathbf{H})} \times 100, \qquad (8)$$

The values of $\mathbf{J_c}'s$ used in the evaluation of diode efficiency are extracted from the results shown in Fig. 2 and Fig. 3, where the onset of resistive states for both polarities of the applied $\mathbf{J}$ is determined. Using this information, in Fig. 4, we present the efficiency of the diode as a function of $\kappa$, for several fixed values $\mathbf{H} \in [1.0, 1.4]$ (in steps of 0.1). For the lowest considered $\mathbf{H} = 1.0$, we observe that the efficiency of the diode reaches its maximum at $\kappa = 2.0$. Around this point, the efficiency is most pronounced, but as the $\kappa$ parameter increases, the efficiency decreases,

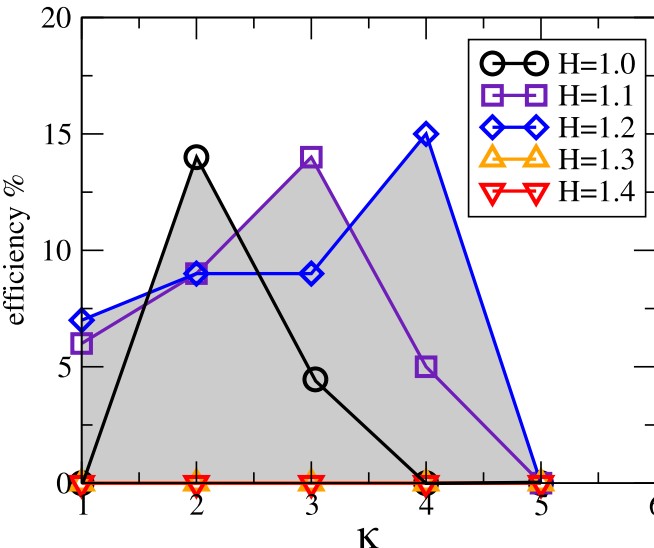

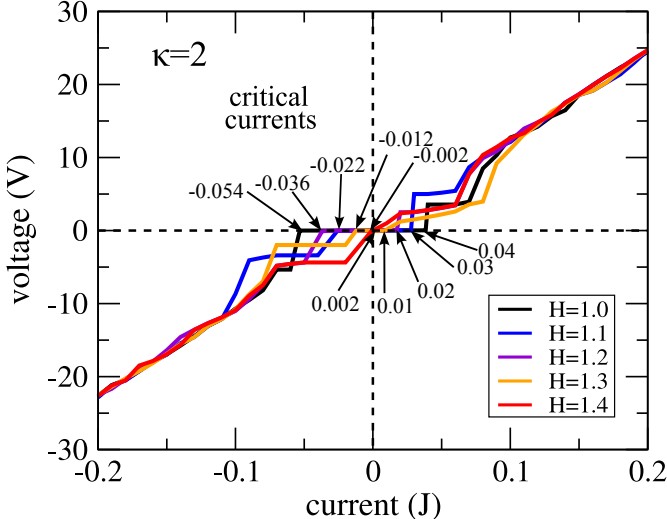

FIG. 4. Diode efficiency $\gamma_d(\mathbf{H})$ as a function of the Ginzburg–Landau parameter $\kappa$, for fixed external magnetic field values $\mathbf{H} \in [1.0, 1.4]$ (in steps of 0.1), in a mesoscopic superconducting wedge acting as a potential diode. As the $\mathbf{H}$ increases, the $\gamma_d(\mathbf{H})$ decreases, and for sufficiently large values of $\mathbf{H}$, the diode effect vanishes, i.e., $\gamma_d(\mathbf{H}) \rightarrow 0$ for all $\kappa$.

FIG. 5. Voltage $V$ as a function of the external current $\mathbf{J}$ for a fixed Ginzburg–Landau parameter $\kappa = 2.0$, considering both current polarities ($\mathbf{J} > 0$ and $\mathbf{J} < 0$), and for different values of the external magnetic field $\mathbf{H} \in [1.0, 1.4]$ (in steps of 0.1).

exhibiting a non-monotonic behavior. This indicates an optimal range of $\kappa$ values where the asymmetry between $\mathbf{J_c}'s$ is maximized. As we increase the external value of $\mathbf{H}$, the position of this maximum efficiency shifts to higher values of $\kappa$. This behavior suggests that stronger $\mathbf{H}$'s tend to favor rectification in samples with reduced superconducting rigidity (larger $\kappa$), possibly due to enhanced vortex mobility and a softer vortex lattice. Nevertheless, regardless of the $\mathbf{H}$ strength or the precise value of the $\kappa$ parameter, the efficiency remains bounded and does not exceed approximately 15%. Furthermore, for very low or very high values of $\kappa$, the efficiency of the diode tends to vanish, as the difference between the $\mathbf{J_c}'s$ for positive and negative $\mathbf{J}$ directions becomes negligible. This suppression of the diode effect occurs even though the meso-wedge geometry breaks the spatial inversion symmetry of the system [16]. This highlights the crucial role played by the interplay between superconducting rigidity ($\kappa$) and vortex dynamics in enabling rectification. Consistent with the previous analysis, the observed maxima in diode efficiency arise from the competition between $\mathbf{J}$ and the energy barriers imposed by the superconducting meso-wedge boundaries. At specific current amplitudes, the vortex entry and motion become strongly direction-dependent due to these asymmetric boundary conditions, thereby enhancing the nonreciprocal response and producing the efficiency peaks. Hence, the maxima reflect optimal operating conditions where the symmetry breaking induced by $\mathbf{J}$ is most effective.

In Fig. 5, we illustrate $V$ as a function of $\mathbf{J}$ for a fixed value $\kappa = 2.0$, and for several values of $\mathbf{H} \in [1.0, 1.4]$ (in steps of 0.1). The results are shown for both polarities:

$\mathbf{J} > 0$ and $\mathbf{J} < 0$. As $\mathbf{H}$ increases, vortex nucleation and penetration into the superconducting region become more favorable, leading to a earlier onset of the resistive state. Our results confirm this: $\mathbf{J_c}$ decreases with increasing $\mathbf{H}$, and the dissipation threshold shifts to lower values of $\mathbf{J}$. Moreover, a clear asymmetry is observed between $\mathbf{J_c}$ for opposite polarities of $\mathbf{J}$, indicating the presence of the superconducting diode effect. This asymmetry is more pronounced at lower $\mathbf{H}$'s and progressively weakens as $\mathbf{H}$ increases, consistent with a reduction in diode efficiency. Compared with the results in Fig. 2, it is evident that the $\mathbf{J_c}$'s are systematically lower due to the enhanced vortex dynamics at higher $\mathbf{H}$ strengths. Beyond $\mathbf{J_c}$, following the initial jump $V$ associated with vortex entry, the system enters a regime characterized by an approximately Ohmic (linear) response [17]. These observations suggest that the diode efficiency as a function of $\kappa$ is likely to exhibit a non-monotonic and nonlinear dependence for varying $\mathbf{H}$ strengths, as illustrated in the case of $\kappa = 2.0$.

With the previous results, in Fig. 6, we present the first $\mathbf{J_c}$ values as a function of $\mathbf{H}$, for a fixed value of $\kappa = 2.0$. The selected $\mathbf{H}$'s are all above the first critical field $\mathbf{H_1}$, where vortices are expected to nucleate and penetrate the superconductor. In this regime, the vortex configurations can differ between $\mathbf{J} > 0$ and $\mathbf{J} < 0$ polarities, potentially leading to distinct $\mathbf{J_c}$ values a key signature of the diode effect [18–20]. The results show a non-monotonic dependence of the $\mathbf{J_c}$ on the $\mathbf{H}$. Notably, the largest asymmetry occurs at $\mathbf{H} = 1.0$, while the $\mathbf{J_c}$'s converge and become equal at $\mathbf{H} = 1.4$ for $\kappa = 2.0$. In between, the $\mathbf{J_c}$'s exhibit alternating increases and decreases, indicating a complex interplay between vortex dynamics and $\mathbf{H}$ strength. This behavior is consistent with the V-$\mathbf{J}$ characteristics shown in Fig. 5. It suggests that the

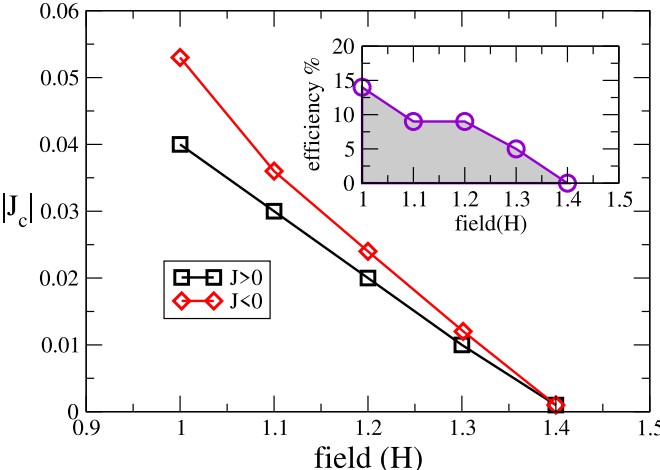

FIG. 6. Magnitude of the first critical currents $\mathbf{J}_c$, for both polarities, as a function of the fixed external magnetic field $\mathbf{H}$ for a fixed Ginzburg-Landau parameter ($\kappa = 2.0$) in the superconducting meso-wedge sample. Inset: efficiency value ($\gamma_d(\mathbf{H})\%$), for both polarities ($\mathbf{J} > 0$ and $\mathbf{J} > 0$), as a function of the fixed external magnetic field $\mathbf{H}$ for a fixed $\kappa = 2.0$ in the superconducting meso-wedge sample.

diode efficiency, as a function of $\kappa$, is expected to exhibit oscillatory or non-monotonic behavior across different values of the $\mathbf{H}$. In the inset of Fig. 6, we show $\gamma_d(\mathbf{H})$ for a fixed $\kappa = 2.0$, as a function of $\mathbf{H} \in [1.0, 1.4]$ (in steps of 0.1). These results are consistent with the analysis of the $\mathbf{J}_c$'s discussed in the previous paragraphs. In particular, $\gamma_d(\mathbf{H})$ exhibits an oscillatory behavior with varying $\mathbf{H}$, reaching a maximum at $\mathbf{H} = 1.0$ and decreasing for larger values of $\mathbf{H}$. Interestingly, despite the presence of vortices in the sample, expected for all selected values $\mathbf{H}$ above $\mathbf{H}_1$, $\gamma_d(\mathbf{H})$ remains non-zero. This suggests that the presence of vortices does not completely suppress the rectification effect and that specific vortex configurations can still lead to a measurable asymmetry between positive and negative $\mathbf{J}_c$'s. These findings indicate that $\gamma_d(\mathbf{H})$ of the superconducting meso-wedge geometry is highly sensitive to the interplay between the external $\mathbf{H}$, $\kappa$, and the geometric asymmetry of the sample. Importantly, this result supports the idea that the diode effect can arise solely from spatial symmetry breaking, without the need for Josephson junctions, as has been commonly proposed in previous studies [46, 47].

With this in mind, we conclude the presentation and analysis of the electronic transport properties for the superconducting meso-wedge. However, the observed asymmetries in the critical currents and diode efficiency suggest that the underlying vortex configurations play a central role in enabling the rectification effect. To explore this hypothesis further, we examine the spatial distribution of the Cooper pair density in the following section, which provides insight into the vortex dynamics and their correlation with the diode effect.

## B. Vortex configuration

The intensity of the order parameter $\psi$ in GL theory represents the macroscopic wave function of the superconducting state, and its squared modulus $|\psi|^2$ corresponds to the local density of Cooper pairs. Spatial variations in $|\psi|^2$ are used to visualize the presence and distribution of vortices, which appear as localized regions of the suppressed $|\psi|^2$. We focus on the spatial distribution of $\psi$ to investigate the role of vortex configurations in the emergence of the diode effect in the superconducting meso-wedge.

In line with this, Fig. 7 shows the colormap of $|\psi|^2$ for a fixed $\mathbf{H} = 1.0$ and two representative values of the GL parameter, in (a) $\kappa = 2.0$ and (b) $\kappa = 4.0$. The plots correspond to two selected $n$ layers along the $\hat{z}$ direction, chosen to highlight differences in vortex behavior. In particular, the lowest layer is shown because it is the one where vortices first penetrate. Fig. 7(a) shows the vortex configurations for $\kappa = 2.0$ in (i) $n = 1$ and (ii) $n = 4$, under both $\mathbf{J} > 0$ and $\mathbf{J} < 0$. For $n = 1$, a clear asymmetry arises between the two $\mathbf{J}$ directions: under $\mathbf{J} < 0$, four vortices are formed (indicated by the black arrow), whereas this configuration is absent for $\mathbf{J} > 0$. For $n = 4$, a similar asymmetry is observed, with two vortices emerging for $\mathbf{J} < 0$, while no such formation occurs for $\mathbf{J} > 0$. These asymmetric vortex patterns are consistent with the asymmetry in the $\mathbf{J}$'s values reported in Figs. 2 and 3, which reveal the enhanced efficiency of the diode effect. In addition, in Fig. 7(b), we consider $\kappa = 4.0$. In contrast to the behavior at lower $\kappa$, the vortex configurations for both layers $n = 1$ and $n = 4$ remain identical under $\mathbf{J} > 0$ and $\mathbf{J} < 0$. This invariance indicates that, at larger $\kappa$, the superconducting meso-wedge no longer exhibits current–direction–dependent asymmetries in the vortex distribution. Correspondingly, at $\kappa = 4.0$, the $\mathbf{J}_c$ values shown in Figs. 2 and 3 are symmetric with respect to $\mathbf{J}$, reflecting a regime where vortex nucleation is independent of $\mathbf{J}$ polarity and governed solely by the magnitude of $\mathbf{H}$. Such behavior is consistent with the expectation that increasing $\kappa$ reduces surface-barrier effects, leading to symmetric vortex entry for opposite $\mathbf{J}$ directions.

Furthermore, Fig. 8 shows the colormap of $|\psi|^2$ for the superconducting meso-wedge under (a) $\mathbf{H} = 1.1$ and (b) $\mathbf{H} = 1.3$, keeping $\kappa = 2.0$ for two representative layers: (i) $n = 1$ and (ii) $n = 7$. In Fig. 8(a), an asymmetry in vortex nucleation is clearly visible between $\mathbf{J} > 0$ and $\mathbf{J} < 0$ for both $n = 1$ and $n = 7$. For example, at $n = 1$ and $\mathbf{J} > 0$, two well-defined vortices are observed (black arrows), while for $\mathbf{J} < 0$ four vortices appear. At $n = 7$, the number of vortices is the same for both $\mathbf{J}$ directions, but the nucleation pattern differs significantly. In Fig. 8(b), corresponding to $\mathbf{H} = 1.3$, asymmetry is again present in both layers. For $n = 1$, for example, one vortex is nucleated under $\mathbf{J} > 0$ (black arrow), while two vortices emerge for $\mathbf{J} < 0$. Similarly, at $n = 7$, one vortex appears for $\mathbf{J} > 0$, whereas three vortices are present for $\mathbf{J} < 0$. From the perspective of transport efficiency, the critical $\mathbf{J}$ data in Figs. 5 and 6 demonstrate a diode

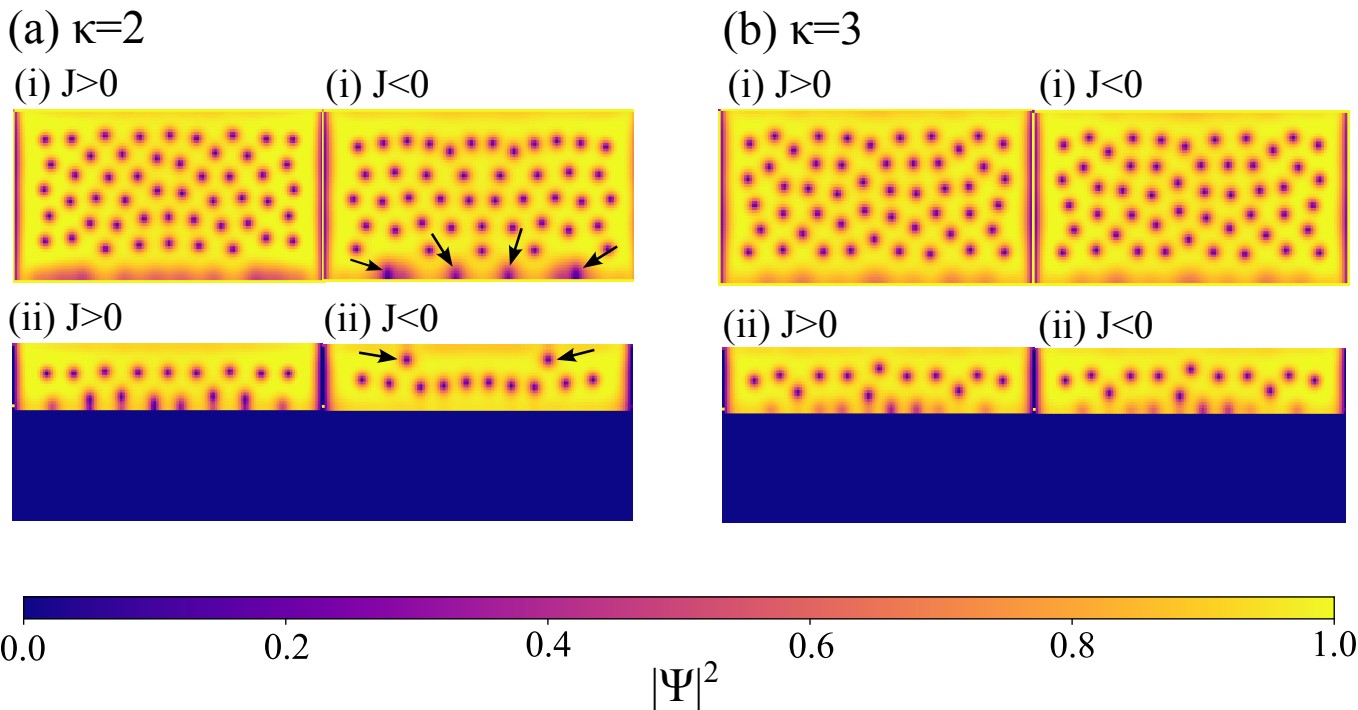

FIG. 7. Color map of Cooper pair density $|\psi|^2$ for different values of the Ginzburg–Landau parameter GLP $\kappa$. Panel (a): $\kappa = 2.0$ and panel (b): $\kappa = 4.0$. The external magnetic field is set to $\mathbf{H} = 1.0$. The vortex states are projected onto three different layers: (i) $n = 1$, and (ii) $n = 4$. The color bar indicates the intensity of $|\psi|^2$ across the superconducting meso-wedge geometry.

effect, confirming a direct correlation between asymmetric Abrikosov vortex nucleation and the emergence of the superconducting diode effect in the meso-wedge geometry.

Having established the theoretical framework, we now focus on the experimental observations that validate and illustrate the physical behavior of the superconducting meso-wedge.

### C. Discussion and experimental evidences

There is experimental support for the presence and role of vortices in the emergence of the superconducting diode effect. Although imaging and resolving vortex configurations under specific values of the external magnetic field $\mathbf{H}$ and applied current $\mathbf{J}$ can be technically challenging, recent advances—particularly the development of nanoscale SQUID-on-tip (SOT) microscopy have enabled direct measurements of local magnetic flux and vortex distributions in superconducting systems. One of the most detailed experimental studies in this context is presented by A. Gutfreund et al. [65], who investigate Nb/EuS (S/F) bilayers and measure the voltage $V$ as a function of the external applied current $\mathbf{J}$ for a fixed $\kappa$. Remarkably, their data exhibit distinct Shapiro steps—quantized voltage plateaus that result from phase locking between the time-dependent superconducting order parameter and an external frequency scale, such as the motion of vortices or internal Josephson oscillations.

The appearance of these steps in their measurements closely resembles our theoretical predictions shown in Figs. 2 and 5, where similar features arise from dynamic vortex entry and collective motion under increasing $\mathbf{J}$. In addition, Gutfreund et al. report pronounced asymmetries in the critical $\mathbf{J}$ values for opposite polarities of $\mathbf{J}$ ($\mathbf{J} > 0$ and $\mathbf{J} < 0$), consistent with the diode-like behavior predicted in our model (Figs. 3 and 6). Importantly, their imaging of vortex configurations reveals non-trivial spatial arrangements: instead of forming regular Abrikosov triangular lattices, vortices are found to align along the sample boundaries or exhibit irregular clustering patterns. These features are shaped by the combined effects of geometric confinement, surface energy barriers, and the Lorentz force acting on vortices under transport $\mathbf{J}$'s. Strikingly similar patterns emerge in our theoretical $|\psi|^2$ maps (Figs. 7 and 8), particularly near asymmetric boundaries, where vortex nucleation and trapping are strongly geometry-dependent. Such *vortex asymmetries* is critical for manifesting the diode effect: rectification arises when the vortex dynamics—specifically their nucleation sites, mobility, and paths—differ under current reversal. No net diode effect would be observed if vortex motion were symmetric for both directions of $\mathbf{J}$. Therefore, the close agreement between vortex configurations and critical $\mathbf{J}$ asymmetries in both our theoretical model and the experimental findings of Gutfreund et al. provides strong evidence that vortex-mediated mechanisms are central to the non-reciprocal transport observed in superconducting diode systems.

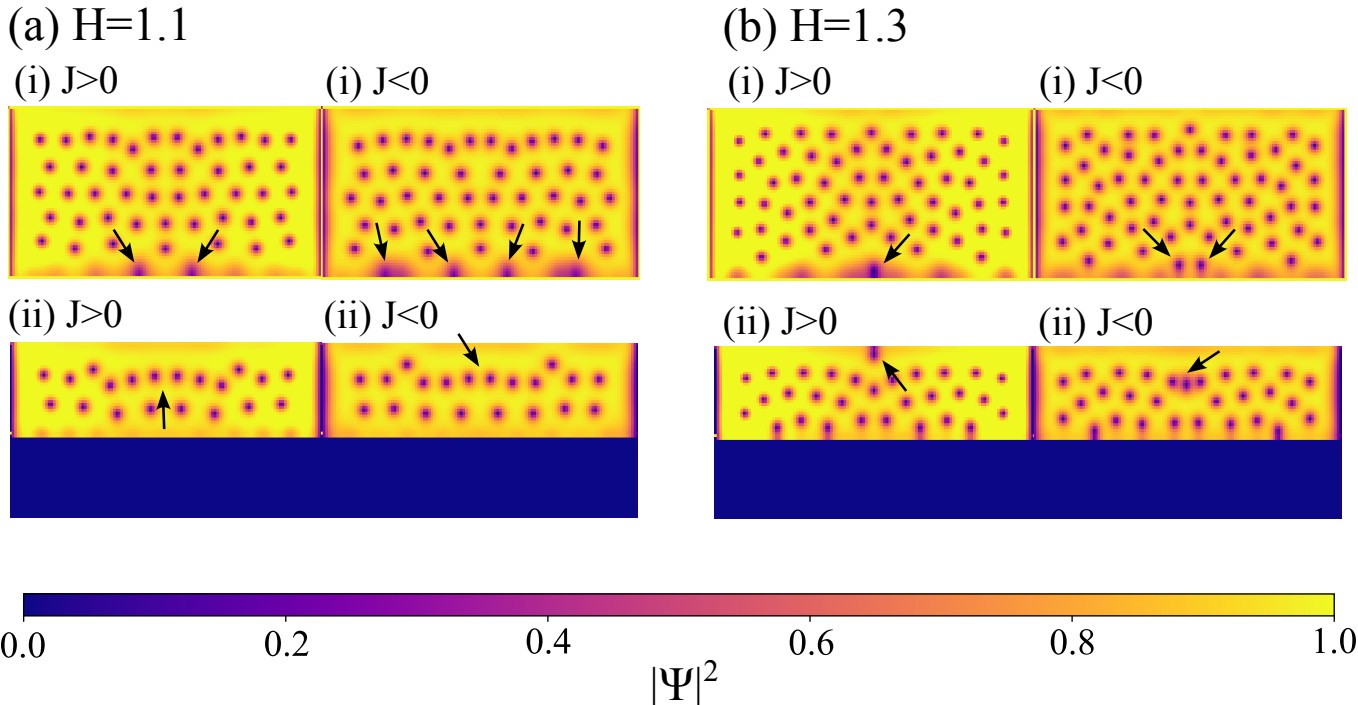

FIG. 8. Cooper pair density $|\psi|^2$ for different values of the external magnetic field **H**. Panel (a): **H** = 1.1, and panel (b): **H** = 1.3. The Ginzburg–Landau parameter (GLP) is set to $\kappa = 2.0$. The density is projected for two selected layers of the superconducting meso-wedge: $n = 1$, and $n = 7$. The color bar indicates the intensity of $|\psi|^2$ in each region of the superconducting meso-wedge geometry.

Castellani *et al.* [66] explores the diode effect in superconducting niobium nitride micro-bridges in a complementary work. They measure the critical **J**'s and diode efficiency. Their results reveal a peak in efficiency at specific **H** and **J** combinations, which follows a functional dependence remarkably similar to that obtained in our simulations (Fig. 4 and Fig. 6, inset). This supports the hypothesis that diode efficiency is maximized when vortex configurations are strongly asymmetric and minimized when the system approaches dynamical symmetry. Also, Taras *et al.* [67] investigate the superconducting diode effect from the perspective of *nonreciprocity induced by spatial symmetry breaking* in a conventional Nb superconductor. Notably, their experiments demonstrate that even without an external magnetic field, geometric asymmetries in the sample can lead to rectification of the applied **J**. They show that the difference between forward and reverse critical **J**'s persists up to zero temperature, an observation that matches the predictions of our theoretical framework in the limit of low thermal fluctuations and dominant geometric effects.

These experimental studies validate the formalism and results presented in this work. The convergence between theory and experiment—regarding vortex distributions, critical current asymmetries, Shapiro steps, and diode efficiency—highlights the key role of mesoscopic geometry and vortex dynamics in the superconducting diode effect. Our results suggest that nonreciprocal superconducting behavior does not necessarily require Josephson junctions or spin-orbit coupling, but can emerge naturally from the interplay between vortex physics and broken spatial symmetry.

From a theoretical point of view, a further possibility to enhance diode efficiency would be to introduce controlled anisotropies—either intrinsic to the crystal lattice, engineered through artificial pinning patterns, or induced geometrically—so that vortex motion becomes directionally dependent, thus reinforcing rectification. In particular, optimizing the wedge angle, tuning the orientation and strength of the applied magnetic field, or introducing a controlled gradient of defects or thermal inhomogeneities could further amplify the asymmetry in vortex dynamics and lead to higher diode efficiencies.

## IV. CONCLUSIONS

In this work, we have theoretically investigated the superconducting diode effect in a superconducting meso-wedge, focusing on the influence of the Ginzburg–Landau parameter $\kappa$ and the external magnetic field **H** on the resistive state and vortex dynamics. By computing the critical current $\mathbf{J}_c$ for both transport current **J** polarities (**J** > 0 and **J** < 0), we identified the emergence of a diode effect characterized by asymmetric transport, quantified via an efficiency parameter $\gamma_d(\mathbf{H})$. Notably, $\gamma_d(\mathbf{H})$ exhibits a non-monotonic dependence on $\kappa$, reaching a maximum at intermediate values and vanishing for larger $\kappa$, suggesting a relationship with the

rigidity of the superconducting condensate and the interplay between vortex mobility and energy barriers.

Our simulations reveal that the diode effect persists even in Abrikosov vortices and does not rely on Josephson junctions or externally imposed symmetry breaking. Instead, it emerges intrinsically from the spatial asymmetry of the sample geometry and the vortex dynamics it induces. By analyzing the Cooper pair density $|\psi|^2$ across multiple layers of the meso-wedge, we show that vortex nucleation is highly sensitive to both geometry and the value of $\kappa$, resulting in nontrivial configurations that break inversion symmetry and differ between opposite current directions. The observed asymmetry in vortex configurations for $\mathbf{J} > 0$ and $\mathbf{J} < 0$ serves as a direct microscopic signature of the superconducting diode effect. In particular, vortex entry preferentially occurs near thinner regions of the sample, where reduced material thickness lowers the energy barrier, reinforcing the directional vortex motion under applied current.

Furthermore, we observe that increasing the magnetic field modifies the spatial distribution of vortices and suppresses $\gamma_d(\mathbf{H})$, indicating a competition between $\mathbf{H}$ strength and geometric confinement. Our results demonstrate that the superconducting diode effect can arise purely from vortex-mediated mechanisms and broken spatial symmetry, offering a novel and intrinsic route to nonreciprocal superconducting transport. These findings align closely with recent experimental observations based on SQUID-on-tip microscopy and critical $\mathbf{J}$ measurements, supporting the relevance and validity of the proposed theoretical framework. Overall, our work contributes to the fundamental understanding of superconducting rectification and may guide the design of future nonreciprocal superconducting devices without relying on complex hetero-structures or artificial junctions.

## V. ACKNOWLEDGMENTS

C. Aguirre wants to thank S. Aguirre and M. Aguirre for useful discussions. C. Aguirre thanks the CNPq grant number process: 174045/2023-9 for financial support. J. Fáundez acknowledges the support from ANID Fondecyt grant number 3240320. Powered@NLHPC: This research was partially supported by the NLHPC supercomputing infrastructure (CCSS210001).

## VI. DATA AVAILABILITY

The code used for the simulations was implemented in **Matlab**® and is available upon request from the corresponding author. The computational time required to obtain results depends on the sample size and simulation parameters. For the sample studied in this work, the average computation time was approximately 12 days, using a workstation equipped with an AMD Ryzen™ 9 7950X3D processor (32 threads), 64 GB of RAM, and running Ubuntu as the primary operating system.

## VII. CONFLICT OF INTERESTS

The authors declare that they have no conflict of interest.

## VIII. AUTHOR CONTRIBUTION

All authors discussed the results and contributed to the final manuscript.

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
