# Peer review of "Superconducting diode effect in a meso-wedge geometry with Abrikosov vortices"

_SciPost Physics Core_

## Round 1 · Referee Report · Anonymous (Referee 1) · 2025-8-6

Strengths

  • Timely problem linking Abrikosov vortices to nonreciprocal superconducting transport
  • Good organisation and presentation of results

Report

Report on "Superconducting diode effect in a meso-wedge geometry with Abrikosov vortices" by C. A. Aguirre , J. Barba-Ortega , A. S. de Arruda and J. Faundez.

The authors of the present manuscript study nonreciprocal superconducting transport in a "meso-wedge" superconducting setup by employing the generalized time-dependent Ginzburg-Landau theory. I find the topic timely and the results interesting, adding an alternative way for diode physics besides systems with spin-orbit coupling and magnetism. For this reason, I think the manuscript warrants publication in Sci. Post. However, before recommending for publication, I think the authors need to improve some parts in relation to the motivation, clarify, novelty, and calculation aspects. For this purpose, I have prepared a list of questions/comments.

  1. In the introduction, the authors start discussing Abrikosov vortices (first paragraph) and the common way to describe them (second paragraph). Then, in the third paragraph, the authors jump to discuss the superconducting diode effect. This third paragraph needs to be better motivated. Why the diode effect in systems with Abrikosov vortices? A better start of this third paragraph would help the reader.

  2. In the third paragraph of the introduction, the authors also mention "spin-orbit interactions that break time-reversal symmetry". This is perhaps a typo. Time-reversal symmetry is broken by magnetic fields, while inversion symmetry can be broken by spin-orbit coupling such as of Rashba type. It is perhaps important to stress the conditions for realizing a diode effect in terms of symmetries, discussed in New J. Phys. 24, 053014 (2022). Here, the authors also mention the diode effect in Josephson junctions but dont cite previous works; see e. g., Phys. Rev. X 12, 041013 (2022); Nature Physics 18, 1228–1233 (2022); Sci. Adv.8,eabo0309(2022); Phys. Rev. B 109, L081405 (2024); Phys. Rev. Lett. 129, 267702 (2022); Phys. Rev. B 111, L140506 (2025), and references therein.

  3. In the third paragraph of the intro, the authors end highlighting systems where the diode effect has been studied. While this is good, the authors need to highlight what is the problem they want to solve. Given that there have also been previous studies on diode physics and vortices [Appl. Phys. Lett. 94, 242504 (2009); Appl. Phys. Lett. 111, 062602 (2017); Phys. Rev. B 100, 174511 (2019); Phys. Rev. Applied 22, 064017 (2024)], it is important that the authors motivate the problem they solve. In particular, why the authors look at the superconducting diode effect in a meso-wedge superconducting system? This needs to be motivated in the third paragraph so that paragraph four is smoothly connected. This way will also help identifying what is the novelty of the presented findings, which, at the moment, are hard to identify.

  4. Following the previous comment, I find it difficult to understand why the authors consider a meso-wedge superconducting system? Why meso-wedge? This needs to be motivated/justified.

  5. In the last paragraph of the intro, the authors say what they want to study. However, it would be more useful if the authors also said what they found.

  6. Section II: While the authors discuss some of the expressions of the time-dependent Ginzburg-Landau theory, it is not clear how it is calculated the superconducting currents that they plot. What is the calculated voltage and how it is found? It has to be clear how the authors calculate the quantities that are later presented in the figures. Also, it is useful to further enhances the physical meaning of \kappa, right after Eqs.(2-4)

  7. In Section II, I also advise the authors to discuss the symmetries broken in their system and the needed symmetries for the diode effect, see New J. Phys. 24, 053014 (2022). This will help understand why the superconducting diode effect is expected.

  8. What is the main message of Fig.2 in relation to the later discussed nonreciprocal transport? Each time the authors show a figure, it is better to motivate the physics behind it so that the reader knows where the story is going.

  9. In Fig. 3: The authors say they plot the first critical current. But, what is the first critical current? Is there a second critical current? Why the first critical current?

  10. În Fig. 4, the authors find maxima of the diode's efficiency. Is there a physical explanation for these maxima? How one can achieve higher efficiencies?

  11. I also strongly advise to improve the presentation and discussion of Figs.7,8. The way they are now is not the best. The authors could consider separating a bit the (a), (b), (c) panels and use arrows to point in the yellow/blue regions when they discuss the physics in the text.

Recommendation

Ask for major revision

  • validity: good
  • significance: good
  • originality: good
  • clarity: good
  • formatting: acceptable
  • grammar: reasonable

Author:  Julian Faundez  on 2025-08-25  [id 5756]

(in reply to Report 1 on 2025-08-06)
Disclosure of Generative AI use

The comment author discloses that the following generative AI tools have been used in the preparation of this comment:

In the MS, we used the free version of DeepL Write solely to improve grammatical coherence and sentence fluency. None of the data, results, or scientific interpretations were generated or modified by artificial intelligence.

Category:
answer to question

We are grateful to the referees for their expert evaluation of our MS.

The authors of the present manuscript study non-reciprocal superconducting transport in a "meso-wedge" superconducting setup by employing the generalized time-dependent Ginzburg-Landau theory. I find the topic timely and the results interesting, adding an alternative way for diode physics besides systems with spin-orbit coupling and magnetism. For this reason, I think the manuscript warrants publication in Sci. Post. However, before recommending for publication, I think the authors need to improve some parts in relation to the motivation, clarify, novelty, and calculation aspects. For this purpose, I have prepared a list of questions/comments.

Q1: In the Introduction, the authors begin to discuss Abrikosov vortices (first paragraph) and the common way to describe them (second paragraph). Then, in the third paragraph, the authors jump to discuss the superconducting diode effect. This third paragraph needs to be better motivated. Why the diode effect in systems with Abrikosov vortices? A better start to this third paragraph would help the reader.

R1: Thank you for calling our attention of this important topic. The clear transition of topics and the specific motivation for the central topic of this paper. In this regard, we have added a paragraph linking the explanation of the diode effect in the presence of Abrikosov-type vortices. The new text was added on page 1, paragraph 3, of the new version of the MS, as:

"In this context, Abrikosov vortices provide a natural platform to explore the superconducting diode effect. Their cores and circulating currents generate strongly inhomogeneous current patterns that are coupled to the geometry and boundaries of the sample, thus introducing an intrinsic directionality in the transport response [21-24]. As a result, mesoscopic superconductors hosting vortices can display distinct critical currents for opposite current directions, making them ideal systems to investigate and enhance diode-like superconducting behavior [25-28]."

Also, four new references have been added as:

[21] Y.\ Wang, M.\ Li, C.\ Pei, L.\ Gao, K.\ Bu, D.\ Wang, X.\ Liu, L.\ Yan, J.\ Qu, Critical current density and vortex phase diagram in the superconductor, \href{https://doi.org/10.1103/PhysRevB.106.054506}{Phys. Rev. B \textbf{106}, 054506 (2022).}

[22] S.\ Kozlov, J.\ Lesueur, D.\ Roditchev, et al., Dynamic metastable vortex states in interacting vortex lines, \href{https://doi.org/10.1038/s42005-024-01645-2}{Commun. Phys. \textbf{7}, 183 (2024).}

[23] R.\ W{\"o}rdenweber, P.\ Dymashevski, V.\ R.\ Misko, Guided vortex motion in superconductors with asymmetric pinning landscapes, \href{https://doi.org/10.1103/PhysRevB.69.184504}{Phys. Rev. B \textbf{69}, 184504 (2004).}

[24] C.\ Reichhardt, C.\ J.\ Olson Reichhardt, Vortex ratchets and diode effects in superconductors with periodic pinning arrays, \href{https://doi.org/10.1103/PhysRevB.78.224511}{Phys. Rev. B \textbf{78}, 224511 (2008).}

[25] G. Carapella; V. Granata; F. Russo; G. Costabile, Bistable Abrikosov vortex diode made of a Py–Nb ferromagnet-superconductor bilayer structure, \href{https://doi.org/10.1063/1.3155424}{Appl. Phys. Lett. \textbf{94}, 242504 (2009).}

[26] Boris Chesca, Daniel John, Richard Pollett, Marat Gaifullin, Jonathan Cox, Christopher J. Mellor, and Sergey Savelev, Magnetic field tunable vortex diode made of $YBa_{2}Cu_{3}O_{7-\delta}$ Josephson junction asymmetrical arrays, \href{https://doi.org/10.1063/1.4997741}{Appl. Phys. Lett. \textbf{111}, 062602 (2017).}

[27] T. Golod, A. Pagliero, and V. M. Krasnov, Two mechanisms of Josephson phase shift generation by an Abrikosov vortex, \href{https://doi.org/10.1103/PhysRevB.100.174511}{Phys. Rev. B \textbf{100}, 174511 (2019).}

[28] Changlong Wang, Guojing Hu, Xiang Ma, Haige Tan, Junjie Wu, Yan Feng, Shasha Wang, Ruimin Li, Bo Zheng, James Jun He, and Bin Xiang Superconducting-diode effect induced by inversion-symmetry breaking in a stepped ${\mathrm{NbSe}}_{2}$ nanoflake, \href{https://link.aps.org/doi/10.1103/PhysRevApplied.22.064017}{Phys. Rev. Applied 22, 064017 (2024).}

Q2: In the third paragraph of the introduction, the authors also mention "spin-orbit interactions that break time-reversal symmetry". This is perhaps a typo. Time-reversal symmetry is broken by magnetic fields, while inversion symmetry can be broken by spin-orbit coupling such as of Rashba type. It is perhaps important to stress the conditions for realizing a diode effect in terms of symmetries, discussed in New J. Phys. 24, 053014 (2022). Here, the authors also mention the diode effect in Josephson junctions but do not cite previous works; see e. g., Phys. Rev. X 12, 041013 (2022); Nature Physics 18, 1228–1233 (2022); Sci. Adv.8,eabo0309(2022); Phys. Rev. B 109, L081405 (2024); Phys. Rev. Lett. 129, 267702 (2022); Phys. Rev. B 111, L140506 (2025), and references therein.

R2: We thank the referee for this important remark. Indeed, in the third paragraph of the introduction the phrase “spin–orbit interactions that break time-reversal symmetry” was a typo. We have corrected it to properly state that time-reversal symmetry is broken by magnetic fields, whereas inversion symmetry can be lifted by spin–orbit coupling of Rashba type. Following the referee’s suggestion, in the revised version we now emphasize more clearly the symmetry requirements for realizing the superconducting diode effect, in agreement with the framework presented in New J. Phys. 24, 053014 (2022). In particular, we highlight that the simultaneous breaking of inversion ($\mathcal{P}$) and time-reversal ($\mathcal{T}$) symmetries is essential to obtain nonreciprocal transport. From this perspective, the meso-wedge geometry represents a natural platform: the triangular confinement intrinsically breaks inversion symmetry, while the presence of Abrikosov vortices under applied magnetic fields explicitly breaks time-reversal symmetry. These cooperative mechanisms not only fulfill the fundamental conditions for superconducting diode behavior, but are also consistent with recent experimental observations of diode effects in wedge-shaped mesoscopic superconductors. A new paragraph was added on page 2, paragraph 2, in the new version of the MS as:

"A central requirement for the realization of the superconducting diode effect is the simultaneous breaking of fundamental symmetries. As emphasized in Ref. [48], nonreciprocal supercurrents emerge only when both inversion symmetry ($\mathcal{P}$) and time-reversal symmetry ($\mathcal{T}$) are broken. Inversion symmetry can be lifted by structural asymmetry or by intrinsic spin–orbit interactions such as those of Rashba type, whereas time-reversal symmetry is broken by the presence of an external magnetic field or magnetic order. The concomitant violation of $\mathcal{P}$ and $\mathcal{T}$ generates magnetochiral anisotropy, which manifests as direction-dependent critical currents, $I^{+}{c}\neq I^{-}{c}$. Within this symmetry-based framework, the meso-wedge geometry provides a natural platform to realize the diode effect: the wedge shape intrinsically breaks inversion symmetry through its triangular confinement, while the penetration of Abrikosov vortices under an applied magnetic field explicitly breaks time-reversal symmetry. The cooperative action of geometry and vortex dynamics therefore creates the fundamental conditions for nonreciprocal superconducting transport, as recently confirmed by experimental observations in wedge-shaped mesoscopic superconductors [49].

Where the new references are:

[48] James Jun He, Yukio Tanaka and Naoto Nagaosa, A phenomenological theory of superconductor diodes, \href{https://doi.org/10.1088/1367-2630/ac6766}{New J. Phys. \textbf{24}, 053014 (2022).}

[49] P. J. Moll, Geometrical design of 3D superconducting diodes., \href{ https://doi.org/10.1038/s43246-025-00788-1}{Commun Mater \textbf{6}, 73 (2025).}}

In addition, we have expanded the introduction by citing the recent advances on the diode effect in Josephson junctions, including:

[32] Y.\ Zhang, Y.\ Gu, P.\ Li, J.\ Hu and K.\ Jiang, General Theory of Josephson Diodes, \href{https://doi.org/10.1103/PhysRevX.12.041013}{Phys. Rev. X, \textbf{12}, 041013 (2022).}

[33] P.\ B.\, Chakraborty, A.\, Sivakumar, P.\ K.\ et al, Josephson diode effect from Cooper pair momentum in a topological semimetal, \href{https://doi.org/10.1038/s41567-022-01699-5}{Nat. Phys. \textbf{18}, 1228–1233 (2022).}

[34] M.\ Davydova, S.\ Prembabu and L.\ Fu, Universal Josephson diode effect, \href{https://doi.org/DOI: 10.1126/sciadv.abo0309}{ Sci. Adv. \textbf{8}, 0309 (2022).}

[35] Jorge Cayao, Naoto Nagaosa, and Yukio Tanaka, Enhancing the Josephson diode effect with Majorana bound states, \href{https://doi.org/10.1103/PhysRevB.109.L081405}{Phys. Rev. B \textbf{109}, L081405 (2024).}

[36] R.S. Souto, M. Leijnse, C. Schrade, Josephson diode effect in supercurrent interferometers, \href{https://doi.org/10.1103/PhysRevB.111.L140506}{Phys. Rev. Lett. \textbf{129}, 267702 (2022).}

[37] A. Costa, O. Kanehira, H. Matsueda, and J. Fabian, Unconventional Josephson supercurrent diode effect induced by chiral spin-orbit coupling, \href{https://doi.org/10.1103/PhysRevB.111.L140506}{Phys. Rev. B \textbf{111}, L140506 (2025).}

We thank again the referee for their meticulous remark.

Q3: In the third paragraph of the intro, the authors end highlighting systems where the diode effect has been studied. While this is good, the authors need to highlight what is the problem they want to solve. Given that there have also been previous studies on diode physics and vortices [Appl. Phys. Lett. 94, 242504 (2009); Appl. Phys. Lett. 111, 062602 (2017); Phys. Rev. B 100, 174511 (2019); Phys. Rev. Applied 22, 064017 (2024)], it is important that the authors motivate the problem they solve. In particular, why the authors look at the superconducting diode effect in a meso-wedge superconducting system? This needs to be motivated in the third paragraph so that paragraph four is smoothly connected. This way will also help identifying what is the novelty of the presented findings, which, at the moment, are hard to identify.

R3: We thank the referee for this comment. In the revised version of the MS, we have added the new text, on page 2, paragraph 3, as:

"Our main goal is understanding how geometric anisotropy influences these system's critical currents, diode efficiency, and vortex nucleation and the influence that the super-current of Abrikosov-type vortices has on critical currents and how the configuration of these vortices is different for different directions of the external applied current".

We also note that part of this point was already addressed in our response to Q2.

Q4: Following the previous comment, I find it difficult to understand why the authors consider a meso-wedge superconducting system? Why meso-wedge? This needs to be motivated/justified.

R4: We thank the referee for this remark, but we note that the justification for using a meso-wedge geometry was already provided in our response to Q2.

Q5: In the last paragraph of the intro, the authors say what they want to study. However, it would be more useful if the authors also said what they found.

R5: We thank the referee for this observation, which we consider has already been addressed in our response to Q3.

Q6: Section II: While the authors discuss some of the expressions of the time-dependent Ginzburg-Landau theory, it is not clear how it is calculated the superconducting currents that they plot. What is the calculated voltage and how it is found? It has to be clear how the authors calculate the quantities that are later presented in the figures. Also, it is useful to further enhances the physical meaning of $\kappa$, right after Eqs.(2-4).

R6: We thank the referee’s suggestion. We added a brief explanation of how the voltage is obtained (from Eq. 4 with our Neumann boundary conditions), clarifying the values reported, for example, in Fig. 2. This new text is in page 2, last paragraph, of the new version of the MS as:

"In the quasi-steady regime, $\partial_t \rho \simeq 0$, so that Laplace’s equation ($\nabla^{2}\Phi=0$) holds in the bulk, with solvability requiring that the net injected current equals the extracted one. Because a pure Neumann problem determines $\Phi$ only up to an additive constant, we fix the reference by enforcing, e.g., $\langle \Phi \rangle_{\partial\Omega_2}=0$. The instantaneous voltage is then computed as the difference of surface-averaged potentials at the electrodes (leads),
\begin{equation}
V(t)=\langle \Phi \rangle_{\partial\Omega_1}-\langle \Phi \rangle_{\partial\Omega_2},
\qquad
\langle \Phi \rangle_{S} = \frac{1}{|S|}\int_{S} \Phi\, dS,
\end{equation}
where, $S$ denotes the leads surface—i.e., a subset of the boundary $\partial\Omega$—over which the potential is averaged. The voltage follows from time-averaging $V(t)$ over the stationary window."

Also, after Eq. (3), we have added a new paragraph discussing the interpretation of $\kappa$ within GL theory, as well as its physical significance. The new text included on page 2, paragraph 1, column 2, after Eq. (3) in the revised version of the MS is:

"and $\kappa$ characterizes the type of superconductor by relating the spatial variation of the order parameter $\psi$ to the magnetic field penetration into the sample. In GL theory, $\kappa$ is a dimensionless quantity defined as the ratio between the magnetic penetration depth and the coherence length of the superconductor. This parameter determines whether a material is classified as type I or type II: for $\kappa < 1/\sqrt{2}$ the superconductor is type I, while for $\kappa > 1/\sqrt{2}$ it is type II. Importantly, we associate the mechanical rigidity of the superconducting vortex lattice with $\kappa$. Larger values of $\kappa$ correspond to a softer vortex lattice, thereby facilitating vortex penetration at lower values of $\mathbf{J}$'s."

We sincerely hope that the above response has addressed and clarified the referee’s concerns, and we thank them once again for their valuable feedback.

Q7: In Section II, I also advise the authors to discuss the symmetries broken in their system and the needed symmetries for the diode effect, see New J. Phys. 24, 053014 (2022). This will help to understand why the superconducting diode effect is expected.

R7: We thank the referee for this useful suggestion. The discussion of the broken symmetries required for the diode effect, in line with New J. Phys. 24, 053014 (2022), has been included in the revised version of the MS. We also note that this point was addressed in our response to Q2.

Q8: What is the main message of Fig.2 in relation to the later discussed nonreciprocal transport? Each time the authors show a figure, it is better to motivate the physics behind it so that the reader knows where the story is going.

R8: We thank the Referee for this helpful remark. The main message of Fig. 2 is to show that the critical currents for positive and negative current directions, $J<0$ and $J>0$, are not identical. This asymmetry is the microscopic origin of the nonreciprocal transport discussed later in the manuscript. In particular, the difference between the two critical currents ($J<0$ and $J>0$) provides the basis for the diode effect, since vortex motion sets in at different thresholds depending on the direction of the applied current $J$.
To make this point clearer, we have included the magnitude of $J$ in Fig. 2 (also Fig. 5) highlighting this difference explicitly. Moreover, this asymmetry is further quantified and discussed in Fig. 3 (also Fig. 6), where we show the extracted values of the first critical currents for both current directions. Together, these results motivate the subsequent analysis of nonreciprocal transport presented in the MS.

Q9: In Fig. 3: The authors they plot the first critical current. But, what is the first critical current? Is there a second critical current? Why the first critical current?.

R9: We thank the Referee for this pertinent question. In Fig. 3 we plot the first critical current $I_{c1}$, which corresponds to the threshold current above which vortices (kinematic or Abrikosov) are depinned and start to move across the sample. At this point, the Lorentz force exceeds the pinning force, leading to vortex motion, dissipation, and a measurable change in the scalar potential. This regime is therefore identified as the onset of the resistive state in the mixed phase of the superconductor.
For completeness, we note that a second critical current $I_{c2}$ also exists. However, this regime is not considered in our study, since for $I>I_{c2}$ the superconducting state is completely destroyed due to depairing, and the system enters the normal state. As our work focuses on vortex dynamics within the superconducting phase, we restrict our analysis exclusively to the first critical current.

Q10: In Fig. 4, the authors find maxima of the diode's efficiency. Is there a physical explanation for these maxima? How one can achieve higher efficiencies?.

R10: We thank the Referee for this important question. The maxima in the diode efficiency reported in Fig. 4 originate from the interplay between the applied supercurrent and the energy barriers at the sample boundaries. When the driving current reaches values where vortex entry and motion are strongly rectified by these boundary conditions, the nonreciprocal response is enhanced, leading to the observed peaks in efficiency. In other words, the maxima correspond to optimal regimes where the breaking of current symmetry is most pronounced.
Concerning the possibility of achieving higher efficiencies, we note that the values obtained for our geometry are already significant. Nevertheless, further enhancement could be achieved by tailoring the pinning landscape inside the superconductor. The controlled introduction of pinning centers or artificial defects would provide additional anchor points for vortices, thereby strengthening rectification. Likewise, optimizing the sample geometry or tuning external parameters such as temperature and magnetic field could shift the system closer to conditions where diode efficiency is maximized.

Q11: I also strongly advise to improve the presentation and discussion of Figs.7,8. The way they are now is not the best. The authors could consider separating a bit the (a), (b), (c) panels and use arrows to point in the yellow/blue regions when they discuss the physics in the text.

R11: We appreciate the referee's comment. Both figures have been improved, and an explanation of the physical phenomenon has been added, based on the evidence in the figures. The new text of the new version of the MS is in page 6, subsection "vortex configuration", paragraphs 1,2 and 3, as:

"The intensity of the order parameter $\psi$ in GL theory represents the macroscopic wave function of the superconducting state, and its squared modulus $|\psi|^2$ corresponds to the local density of Cooper pairs. Spatial variations in $|\psi|^2$ are used to visualize the presence and distribution of vortices, which appear as localized regions of the suppressed $|\psi|^2$. We focus on the spatial distribution of $\psi$ to investigate the role of vortex configurations in the emergence of the diode effect in the superconducting meso-wedge.

In line with this, Fig. (7) shows the colormap of $|\psi|^2$ for a fixed $\mathbf{H}=1.0$ and two representative values of the GL parameter, in (a) $\kappa = 2.0$ and (b) $\kappa = 4.0$. The plots correspond to two selected $n$ layers along the $\hat{z}$ direction, chosen to highlight differences in vortex behavior. In particular, the lowest layer is shown because it is the one where vortices first penetrate. Fig. (7)(a) shows the vortex configurations for $\kappa = 2.0$ in (i) $n=1$ and (ii) $n=4$, under both $\mathbf{J}>0$ and $\mathbf{J}<0$. For $n=1$, a clear asymmetry arises between the two $\mathbf{J}$ directions: under $\mathbf{J}<0$, four vortices are formed (indicated by the black arrow), whereas this configuration is absent for $\mathbf{J}>0$. For $n=4$, a similar asymmetry is observed, with two vortices emerging for $\mathbf{J}<0$, while no such formation occurs for $\mathbf{J}>0$. These asymmetric vortex patterns are consistent with the asymmetry in the $\mathbf{J}$'s values reported in Figs. (2) and (3), which reveal the enhanced efficiency of the diode effect. In addition, in Fig. (7)(b), we consider $\kappa = 4.0$. In contrast to the behavior at lower $\kappa$, the vortex configurations for both layers $n=1$ and $n=4$ remain identical under $\mathbf{J}>0$ and $\mathbf{J}<0$. This invariance indicates that, at larger $\kappa$, the superconducting meso-wedge no longer exhibits current–direction–dependent asymmetries in the vortex distribution. Correspondingly, at $\kappa=4.0$, the $\mathbf{J}_{c}$ values shown in Figs. (2) and (3) are symmetric with respect to $\mathbf{J}$, reflecting a regime where vortex nucleation is independent of $\mathbf{J}$ polarity and governed solely by the magnitude of $\mathbf{H}$. Such behavior is consistent with the expectation that increasing $\kappa$ reduces surface-barrier effects, leading to symmetric vortex entry for opposite $\mathbf{J}$ directions.

Furthermore, Fig. (8) shows the colormap of $|\psi|^{2}$ for the superconducting meso-wedge under (a) $\mathbf{H}=1.1$ and (b) $\mathbf{H}=1.3$, keeping $\kappa = 2.0$ for two representative layers: (i) $n=1$ and (ii) $n=7$. In Fig. (8)(a), an asymmetry in vortex nucleation is clearly visible between $\mathbf{J}>0$ and $\mathbf{J}<0$ for both $n=1$ and $n=7$. For example, at $n=1$ and $\mathbf{J}>0$, two well-defined vortices are observed (black arrows), while for $\mathbf{J}<0$ four vortices appear. At $n=7$, the number of vortices is the same for both $\mathbf{J}$ directions, but the nucleation pattern differs significantly.
In Fig. (8)(b), corresponding to $\mathbf{H}=1.3$, asymmetry is again present in both layers. For $n=1$, for example, one vortex is nucleated under $\mathbf{J}>0$ (black arrow), while two vortices emerge for $\mathbf{J}<0$. Similarly, at $n=7$, one vortex appears for $\mathbf{J}>0$, whereas three vortices are present for $\mathbf{J}<0$. From the perspective of transport efficiency, the critical $\mathbf{J}$ data in Figs. (5) and (6) demonstrate a diode effect, confirming a direct correlation between asymmetric Abrikosov vortex nucleation and the emergence of the superconducting diode effect in the meso-wedge geometry."

We sincerely thank the referee for the thoughtful and constructive suggestions, which have helped us improve the clarity and quality of the MS.

Attachment:

Cover_letterList_changes.pdf

---

## Round 2 · Referee Report · Anonymous (Referee 1) · 2025-10-1

Report

Second report on "Superconducting diode effect in a meso-wedge geometry with Abrikosov vortices" by C. A. Aguirre , J. Barba-Ortega , A. S. de Arruda and J. Faundez.

The authors responded appropriately to most of my comments and think the manuscript has considerably improved. For these reasons I recommend the acceptance of the manuscript.

I would like to note that some of my previous comments, although responded, were not implemented in the manuscript. I believe that such comments are not critical but they can help improve further the manuscript.

  1. In my previous question Q5, I asked the authors to write down their findings in the last paragraph of the introduction. However, they did not address it. Although this is not critical, I think it is useful to discuss the findings in the suggested paragraph, not just mentioning the goals of the work.

  2. I advise the authors to add in their manuscript the explanation that they wrote in their previous response to Q8, all in relation to the discussion of Fig.2.

  3. The same advice as in point 2 above applies for Fig.3 and their response to my previous question Q9.

  4. The response given to my comment Q10 would be good to be implemented in the manuscript as well. Discussing how to achieve higher efficiencies would also improve the manuscript.

Recommendation

Ask for minor revision

---

## Round 2 · Author Response

SciPost Physics Core,
Dear Editor,

We thank you for the opportunity to resubmit our manuscript entitled "Superconducting diode effect in a meso-wedge geometry with Abrikosov vortices" (manuscript scipost_202505_00028v1) for consideration in SciPost Physics Core.

In preparing this revised version, we have not only incorporated new content to strengthen the scientific scope of the work but also carefully revised the entire manuscript to enhance its clarity, coherence, and overall readability. All comments and suggestions provided by the referee have been addressed in detail, and we have made substantial efforts to clarify the motivation, methodology, and significance of our results.

We believe that these improvements, together with the additional material included, have considerably strengthened the manuscript. We are therefore confident that the revised version now meets the high standards of SciPost Physics Core, and we are pleased to resubmit it for your kind consideration.

Thank you for your consideration and we look forward to hearing from you.

Sincerely,

J. Faundez, on behalf of the authors.

---

## Round 2 · List of Changes

Summary of Changes

(1) A new text has been added to the page. 1, paragraph 3, of the new version of the MS.

(2) New text in page 2, paragraph 2 was added to explain the results obtained.

(3) Text was added in page 2, paragraph 3 in the new version of the MS.

(4) Additional text was inserted on page 2, following Eq. (3), in the revised version of the MS.

(5) Additional text was added on page 2, last paragraph, in the revised version of the MS.

(6) Figs. (2) and (5) were improved.

(7) Text was re-written in subsection "vortex configuration", page 6, of the new version of the MS.

(8) The quality of Figures 7 and 8 has been enhanced.

(9) 16 additional references have been incorporated and appropriately cited in the MS.

(10) The English language has been thoroughly reviewed throughout the revised version of the manuscript

All changes are highlighted in red color in the revised version of the MS provided at the end of this report, after the point-by-point replay to the referee.

---

## Round 3 · Author Response

Dear Editor,
We thank you for the opportunity to resubmit our manuscript entitled "Superconducting diode effect in a meso-wedge geometry with Abrikosov vortices" (manuscript ID: BV14591) for consideration in SciPost Physics Core.
Following the referee’s valuable comments, we have carefully revised the manuscript and included all the requested clarifications and improvements. In particular, we have expanded the discussion in the Introduction, Results and Discussion and experimental evidences sections and improved the overall clarity and readability of the MS.
We believe that the revised version of the MS addresses all the referee’s concerns and that the manuscript is now significantly improved. We therefore hope that it will be suitable for publication in SciPost Physics Core.
Thank you for your consideration and we look forward to hearing from you.
Sincerely,
J. Faundez, on behalf of the authors.

---

## Round 3 · List of Changes

-
A new text has been added to page 2, column 1, paragraph 3, of the new version of the MS.
-
A new text added on page 3, column 2, section "numerical results" and paragraph 1, of the new version of the MS.
-
New text on page 4, column 1, and paragraph 2, of the new version of the MS.
-
A new text was added on page 5, column 1, and paragraph 5, of the new version of the MS.
-
A new text on page 8, column 2, and paragraph 3, of the new version of the MS.

---

## Round 4 · Author Response

Dear Editor,
We thank you for the opportunity to resubmit our MS entitled Superconducting diode effect in a meso-wedge geometry with Abrikosov vortices". Manuscript reference: $scipost_202505_00028v3$ for consideration in Scipost Physics Core.
Following Referee 2’s valuable comments, we have thoroughly revised the MS and incorporated all requested clarifications and improvements.
We believe that the revised version adequately addresses all concerns raised by Referee 2 and substantially strengthens the presentation of our results. We therefore hope that the MS is now suitable for publication in SciPost Physics Core.
Thank you for your consideration and we look forward to hearing from you.
Sincerely,
J. Faundez, on behalf of the authors.

---

## Round 4 · List of Changes

(2) New paragraphs have been added, as recommended by the referee.
(3) Text regarding dimensions was removed from the captions of Figures 2 and 5.
(4) We have added a new subsection in the new version of the MS.
(5) The English was completely revised.
All changes are highlighted in blue color in the revised version of the MS.

---

## Editorial Decision

editorial_decision: